# Additively manufacturable micro-mechanical logic gates

Yuanping Song [1], Robert M. Panas[2], Samira Chizari[1], Lucas A. Shaw[1], Julie A. Jackson [2], Jonathan B. Hopkins [1] & Andrew J. Pascall [2]

Early examples of computers were almost exclusively based on mechanical devices. Although electronic computers became dominant in the past 60 years, recent advancements in three-dimensional micro-additive manufacturing technology provide new fabrication techniques for complex microstructures which have rekindled research interest in mechanical computations. Here we propose a new digital mechanical computation approach based on additively-manufacturable micro-mechanical logic gates. The proposed mechanical logic gates (i.e., NOT, AND, OR, NAND, and NOR gates) utilize multi-stable micro-flexures that buckle to perform Boolean computations based purely on mechanical forces and displacements with no electronic components. A key benefit of the proposed approach is that such systems can be additively fabricated as embedded parts of microarchitected metamaterials that are capable of interacting mechanically with their surrounding environment while processing and storing digital data internally without requiring electric power.

[1] Department of Mechanical and Aerospace Engineering, University of California, Los Angeles, Los Angeles, CA 90095, USA. [2] Materials Engineering Division, Lawrence Livermore National Laboratory, 7000 East Avenue, Livermore, CA 94550, USA. Correspondence and requests for materials should be addressed to J.B.H. (email: hopkins@seas.ucla.edu) or to A.J.P. (email: pascall1@llnl.gov)

To overcome limitations of semiconductor electronics (e.g., unstable performance and failure in extreme environments[1]), researchers have been seeking alternative computational approaches and logic computing devices on small enough scales to perform sufficiently fast calculations in a compact space. Such micro-scale devices include MEMS (micro-electro-mechanical system) logic gates[2,3] and switches[4], all-optical logic gates[5], microfluidic logic devices based on droplets[6] and bubbles[7], and magnetic bubble logic devices[8]. Logic devices on an even smaller scale have also been proposed and studied, such as molecular logic gates[9] and quantum logic gates[10]. A major advantage of micro-mechanical logic devices[11,12] is that they utilize energy in a mechanical form and require no electrical power source or electronic components. As a result, such devices generate no electromagnetic signature and are highly insensitive to radiation damage. Furthermore, studies on reversible computing have suggested that mechanical logic systems can theoretically be designed such that the energy dissipation in the system can be arbitrarily small[13,14] (i.e., the computations consume nearly zero energy). Several groups of researchers have begun to explore the feasibility of mechanical computing systems. Merkle et al[15]. proposed a Turing-complete[16] mechanical computing system based on rigid links and rotary joints. The system operates under a clocked external excitation, and the digital signal is represented by the displacement of rigid links that are guided and constrained by two basic mechanisms named locks and balances. Ion et al.[17] designed a functional-complete[18] mechanical logic system that is integrated with 3D-printed metamaterial mechanisms. The digital signal propagates as mechanical impulses between adjacent cells through the embedded bi-stable springs. The bi-stable spring amplifies the incoming signal in a manner analogous to the domino effect and requires a manual reset after each calculation. Raney et al.[19] proposed a structure that propagates mechanical signals over arbitrary distances without distortion. Logic operators, such as the AND gate, the OR gate, and diode logic were demonstrated.

In this paper, we introduce micro-mechanical logic gate designs of AND, OR, NOT, NAND, and NOR. The operations of these logic gates are designed in 2D (two-dimensions) based on multi-stable mechanisms composed of buckling micro-flexures. The chief difference between the designs proposed in this paper and the existing mechanical logic gates is that the proposed designs achieve the following properties simultaneously: Functional completeness: All possible digital logic operations can be expressed by combining the designed logic gates. The functionally complete sets of binary logic gates are {AND, NOT}, {OR, NOT}, {NAND}, and {NOR}, which have all been demonstrated in this paper. Continuous operation: The proposed logic gates do not need to be reset to an initial state prior to the next logic operation. The presence of the input(s) will immediately trigger the operation. Scalable design: The proposed designs can function at all scales including the micro-scale. The flexure-based designs avoid sliding contact between surfaces and therefore avoid hysteresis and failure due to friction and wear[20]. Designs that use sliding-contact bearings are not easy to fabricate and tend to bind due to intermolecular forces that become dominant on the micro-scale. Constant energy storage across different logic states: Each logic gate stores the same amount of total strain energy at different logic states. This allows for nearly zero-energy operation in theory[14].

## Results

**Logic gate design.** The mechanical logic gates presented in this work are based on bi-stable flexure mechanisms. We begin by describing the basic buckled flexure element and how it represents logic states. The undeformed shape of one such buckled flexure design is shown in Fig. 1a. When the structure is compressed horizontally by a fixed distance $h$ that exceeds the critical buckling threshold, the rigid body marked with S will no longer be stable at the original equilibrium position $d_S = 0$, but will instead rest at either one of the two stable positions, $d_S = -s$ and $d_S = +s$ as shown in Fig. 1b, where the $+s$ position represents a logic 1, and the $-s$ position represents a logic 0. By virtue of symmetry, the two stable states occupy the same energy with an energy barrier in between that must be overcome to transition between the states. This extra energy required for the transition is released from the system after the transition completes, and thus can be used to trigger the transition of the next bi-stable flexure logic element regardless of whether it is from 0 to 1 or from 1 to 0. This is particularly advantageous to a large mechanical computing system because it allows for mechanical digital signals to propagate in a wavelike manner with minimum attenuation. Fig. 1c plots the FEA result of the total potential energy $E_p$ stored in the deformed flexures as a function of the position $d_S$ and the compression distance $h$. The magnitude of the energy barrier can be tuned by adjusting the compression distance $h$, which is an additional design parameter apart from geometric dimensions. When the value of $h$ increases beyond the second critical buckling threshold ($h = 0.163 L$ for the design shown)[21], secondary bifurcations occur in the post-buckling system[21,22], resulting in issues such as multi-mode buckling and mode jumping[23,24] as the mechanism switches between the two logic states. Such undesirable behavior may empirically be avoided by constraining the compression distance $h$ to be less than the critical distance $h_2$ corresponding to the second buckling mode. Therefore, when designing the bi-stable flexures, it is more desirable in general to have a large range between the first and second critical compression distances $[h_1, h_2]$. The design topology shown in Fig. 1a is chosen among several alternatives for its better buckling mode stability. The buckling mode analysis results of the proposed design are compared against an alternative Design B in Fig. 1d, where the rigid bodies are simply connected by a pair of parallel beams. The FEA results suggest that Design A has a more stable first buckling mode and thus constructs a more robust bi-stable system. In addition, Design A generates a larger distance between the two stable positions $d_S = \pm s$, which results in a stronger binary signal. Other bi-stable mechanism designs that were considered are provided in Supplementary Table 1.

By utilizing the bi-stable flexure mechanism as a building block, we constructed larger logic gate mechanisms. In these mechanisms, the rigid bodies are linked through bi-stable flexures and the resulting relationships between their motions are used to represent the basic digital logic operations. For instance, a mechanical NOT gate was designed as shown in Fig. 2a. Its working principle is similar to a mechanical inverter proposed by Merkle[14] shown in Fig. 2b, but instead of flexible beams, the proposed NOT gate uses the bi-stable buckling flexures to provide the negation of the input signal. Under a fixed buckling compression distance $h$ (Fig. 2a), the structure has two stable configurations as shown in Fig. 2c, representing the negation of the input 0 and 1, respectively. Unlike many other mechanical NOT gate designs that use rotational joints to reverse the direction of a mechanical displacement or force, this NOT design uses no rotary connectors and therefore is easier to manufacture additively and does not suffer from issues like energy loss and device failure due to friction and wear. A mechanical OR gate was constructed by joining two sets of the bi-stable flexure mechanism together as shown in Fig. 2d. The vertical positions of the two rigid bodies A and B are controlled externally; whereas, the vertical position of the rigid body C represents the output. Similar

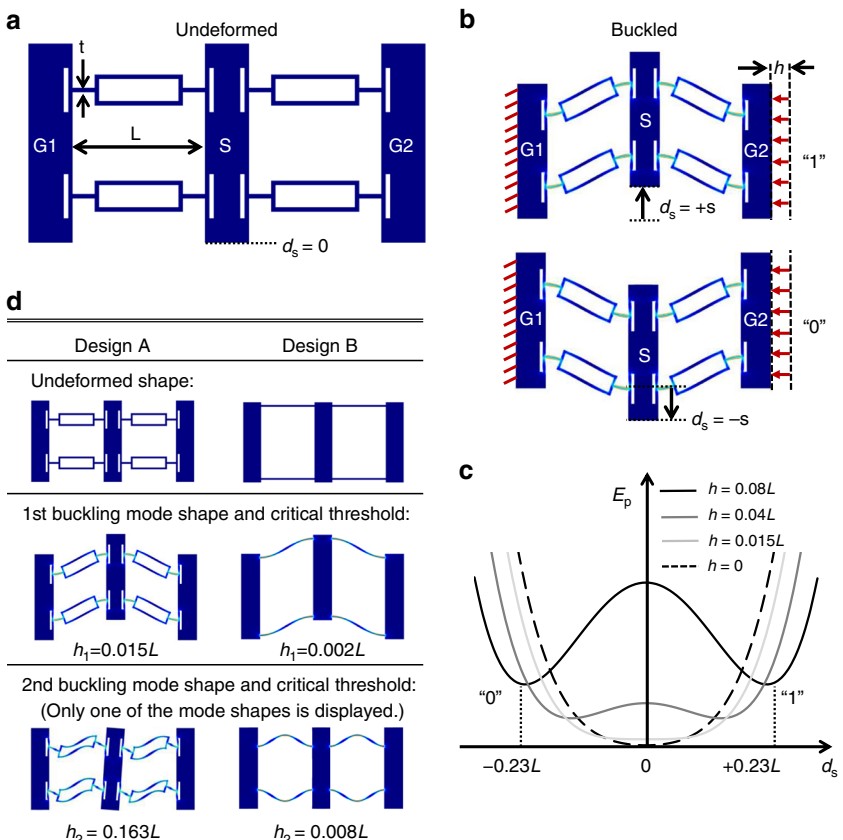

**Fig. 1** Design of bi-stable flexure mechanisms. **a** Undeformed shape of a bi-stable flexure mechanism with a smallest feature size $t = L/30$, where $L$ is the length of a bi-stable flexure. **b** Two buckled shapes of the bi-stable flexure mechanism that are generated using COMSOL Multiphysics® where high-stress regions are indicated by bright colors. **c** Total deformation energy $E_p$ as a function of the position $d_s$ and compression distance $h$ for the bi-stable flexure mechanism in **b**. **d** Undeformed shape, first, and second buckling mode shapes and their corresponding critical compression distance $h_1$ and $h_2$ of two different bi-stable flexure mechanism designs. Design A was chosen for this study due to the large distance between 1st and 2nd buckling modes which means the logic gate is more tolerant to errors in the fixturing during precompression

to the previously described compression process, the OR gate was compressed horizontally by a total distance of 2 $h$ under a set of initial small perturbational forces that resulted in the deformed shape of Mode I in Fig. 2d, where the two inputs are 0 and 0, and the output is 0. When the two inputs A and B switch between logic state 0 and 1, the rigid body C moves accordingly to generate the logic OR operation as shown in Fig. 2d. A functional-complete NAND gate was constructed using the aforementioned OR gate and NOT gates. Inside the NAND gate, the input signals A and B are inverted by two NOT gates, which pass the negation signals to the OR gate through the middle layer X and Y, resulting in a final output C. Figure 2e demonstrates the configurations of the NAND gate at different logic states. The quasi-static timing diagrams of the input(s) and the output of the NOT, OR, and NAND gate were obtained using COMSOL and are shown in Fig. 2f. The FEA result suggests that the proposed logic gates can perform continuous logic operations without resetting to an initial configuration. Similar to the design of the OR and NAND gates, AND and NOR gates were also constructed using the same bi-stable flexure mechanisms, the details of which can be found in Supplementary Note 1.

**Fabrication techniques and experimental results**. A macroscale (about 250 mm by 250 mm) NAND gate was 3D-printed using ABS plastic (with an elastic modulus of 2230 MPa) as shown in Fig. 3a. To deform the structure into the buckled shape, the two rigid bodies G1 and G2 were compressed inward while the rigid

body G3 is fixed to the ground. An Instron mechanical tester was then used to measure the force-displacement relationships of the NAND gate's input(s) when switching between different logic states. Figure 3b plots the total force on the rigid bodies A and B against their vertical displacement when the NAND gate transitioned from Mode I (0 NAND 0) to Mode IV (1 NAND 1). FEA and experimental confirmation of other logic transitions for the NAND and NOR gates can be found in Supplementary Figure 5. The experimental results agree with the FEA results obtained from COMSOL to within the error of the measurement.

A mesoscale (about 25 mm by 25 mm) NAND gate was fabricated using projection microstereolithography[25] as shown in Fig. 3c. In order to fabricate the proposed mechanical logic gates at micro-scale, a new approach[26] is demonstrated that combines the utility of two-photon stereolithography (2PS)[27] with holographic optical tweezers (HOT)[28] into a single apparatus. The 2PS approach is used to print polymer structures with submicron resolution, and the HOT approach is used to exert optical forces on the structure to introduce stored strain energy into the flexures. The complete fabrication and actuation process was demonstrated for a bi-stable buckling flexure mechanism (Fig. 3d). The final fabricated bi-stable element has an overall size of $38 \times 38 \times 3$ μm and flexure thicknesses of 800 nm. The rigid bodies S and G2 were free to move while the rigid body G1 was fixed to the substrate. Next, two optical traps were created at the ends of the rigid body G2 which pulled G2 into contact with G1. This movement caused the flexures to buckle and deform into one of the stable positions. With the two ends of the rigid bars in

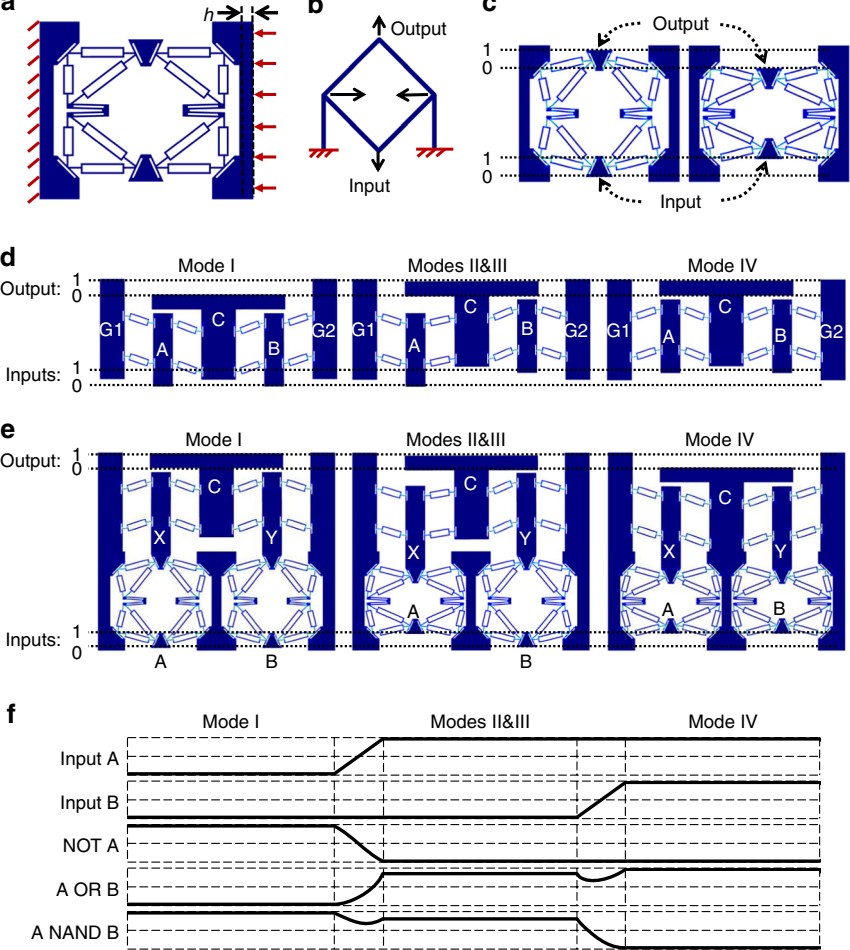

**Fig. 2** Design of mechanical logic gates. **a** Undeformed shape of the NOT gate. **b** A conceptual design of a mechanical inverter using flexible beams. **c** Geometry configurations of the NOT gate at the two stable states representing the negation of logic 0 and logic 1, respectively. **d** Geometry configurations of the OR gate at Mode I (0 OR 0), Modes II & III (0 OR 1 and 1 OR 0), and Mode IV (1 OR 1). **e** Geometry configurations of the NAND gate at Mode I, Modes II & III, and Mode IV. **f** Quasi-static timing diagram of the outputs of NOT, OR, and NAND when the inputs (A, B) transition from (0, 0) to (1, 0) to (1, 1)

contact, they were fused together at the interfaces by locally curing the photopolymer via the 2PS approach. To imitate a digital logic signal, the center bar was driven by an optical trap which switched the bi-stable mechanism between the up position and the down position (Fig. 3e). A video of the fabrication and testing of the micro-scale bi-stable element can be seen in Supplementary Movie 1.

The current 2PS/HOT system has a minimum micro-fabrication resolution of 800 nm and can generate optical trapping forces of up to 50 pN, which in theory is capable of fabricating a logic gate of 100 μm size. A rough estimation from first principles suggests that logic gates at this scale can perform logic operations at Mhz frequencies. This estimation will be further investigated and verified in our future work.

## Discussion

In summary, this paper proposes a digital mechanical computation approach using additively manufacturable micro-mechanical logic gates. Functional logic gates, such as AND, OR, NOT, NAND, and NOR have been designed based on buckling bi-stability and verified using both finite element analysis and experiments. A viable micro-additive manufacture technique has been demonstrated. Our future work includes the development of

three-dimensional (3D) mechanical logic gates and other functional cells, such as memory units and signal amplifiers. Analytical tools will be developed to analyze and optimize buckling bi-stable flexure designs efficiently. The micro-fabrication technique will also be further improved so that a micro-mechanical logic circuit can be demonstrated and tested.

## Methods

**Logic gate fabrication and testing**. The logic gates presented were fabricated using three different methods to demonstrate the ability to additively manufacture them over a range on length scales. The NAND gate was designed in the CAD software Solidworks (Dassault Systèmes) and subsequently exported as an STL file. This STL file was used as the build file for the macro- and mesoscales. The macroscale gate was printed using fused deposition modeling on a Stratasys F370 in ABS (ABS-M30™) and tested on an Instron 5943 mechanical tester using a 3D-printed test fixture to set the desired precompression. The mesoscale gate was printed using projection microstereolithography on a custom-built machine[29] with PR57 resin (Colorado Photopolymer Solutions). The micro-scale bi-stable flexure element was fabricated and tested using a custom built 2PS/HOT system that is described in detailed below. The photopolymer resin used to fabricate this element consists of 1%wt. Li-TPO (Colorado Photopolymer Solutions), 35%wt. ethoxylated (15) trimethylolpropane triacrylate (Sartomer SR9035), and 64%wt. DI water. This photopolymer is specifically designed for the hybrid 2PS/HOT fabrication method as the low viscosity and the difference in refractive index of the cured and uncured polymer enable HOT-based manipulation. First, the CAD model of the bi-stable mechanism (Supplementary Figure 6) was converted into a point cloud which

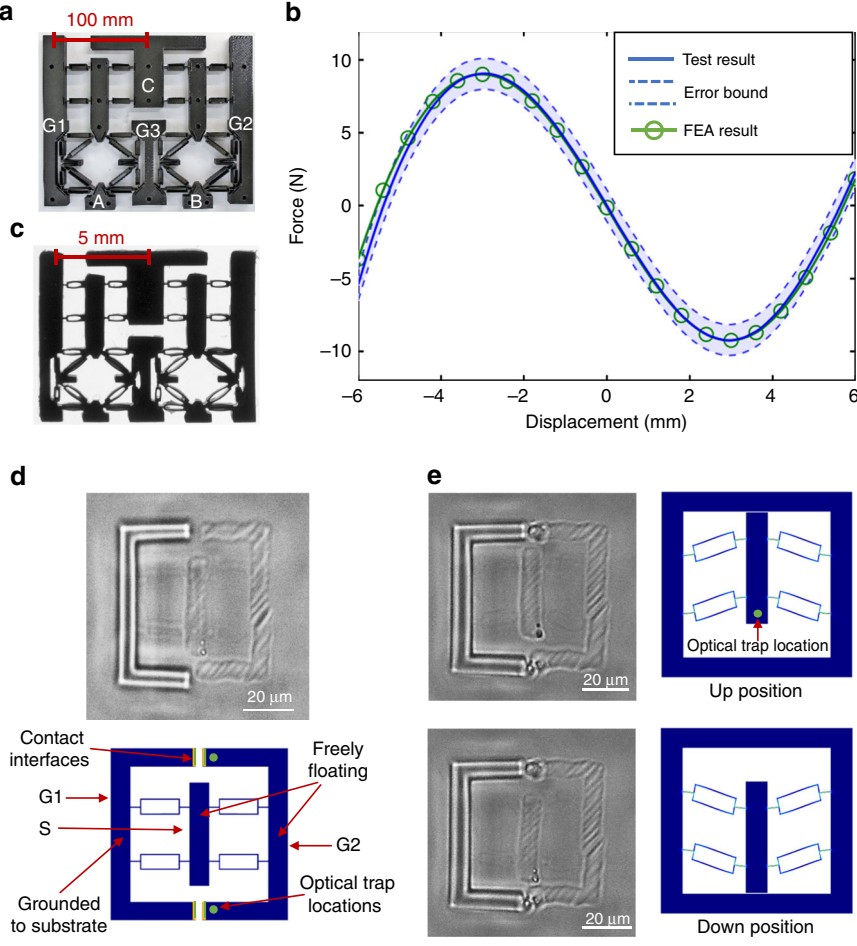

**Fig. 3** Fabrication and experimental testing of the mechanical logic gates. **a** A macroscale NAND gate printed by a commercial Fused Deposition Modeling (FDM) 3D-printer Stratasys F370 using material ABS-M30™. **b** Experimental and FEA results of the total force on the two rigid bodies A and B as the inputs **a**, **b** transition from (0, 0) to (1, 1). **c** A mesoscale NAND gate printed using projection microstereolithography. **d** Fabrication process of a bi-stable flexure mechanism that combines two-photon stereolithography (2PS) with holographic optical tweezers (HOT). **e** A bi-stable flexure mechanism fabricated at micro-scale that can be driven between two stable positions

generated the scan path of a femtosecond pulsed laser in the liquid resin chamber. The undeformed microstructure was then fabricated by the laser at a resolution of 800 nm.

The 2PS system includes a femtosecond laser (fs laser, Spectra-Physics MaiTai eHP DS), acousto-optic modulator (AOM, IntraAction ATM-802DA2 and ME-820–6), two ultrafast mirrors (M), beam block (BB), beam expander system (BE, $f = 50$ and 200 mm lenses), 2D scanning mirror galvanometer (galvo, Thorlabs GVS012), 4-F telescope relay (two $f = 60$ mm lenses), power sensor (Thorlabs S142C and PM100USB), and longpass dichroic mirror (DM2, $\lambda c = 650$ nm). The AOM and 2D scanning mirror galvanometer are driven by an analog output module (National Instruments NI-9263).

The HOT system includes a continuous wave (CW) laser (Laser Quantum Opus 3W at 532 nm), liquid crystal polarization rotator (LCPR, Meadowlark Optics LPR-100–0532), beam block (BB), polarizing beamsplitter (PBS), beam expander (BE1, $f = -50$ and 200 mm lenses), mirror (M), 1920-by-1152 pixel phase-only spatial light modulator (SLM, Meadowlark Optics P1920–0532), 4-F telescope relay ($f = 200$ and 250 mm lenses), Fresnel beam block, and shortpass dichroic mirror (DM1, $\lambda c = 567$ nm).

Both the 2PS and HOT systems use a ×100 oil immersion microscope objective (MO, Olympus Plan Apo Lambda, NA = 1.45) and a three-axis micro-positioning stage (Thorlabs MAX341 and BSC203). The imaging system consists of two cameras (Basler ace acA1300–75 gm) at ×40 and ×100 magnification, doublet lenses (L), beamsplitters (BS1, 50:50 R:T; BS2, 90:10 R:T), tube lens (TL, Thorlabs ITL200), dichroic filters (DF, ND6.0 at 532 nm and 690–1040 nm), and two collimated 617 nm LED illumination sources in both brightfield and darkfield configurations. Supplementary Figure 7 shows a schematic layout of the apparatus.

**Finite element analysis modeling details**. The FEA results of the quasi-static force-displacement relationship were obtained using COMSOL Multiphysics® Structural Mechanics Module. A two-dimensional plane-stress model was

established, and the geometry was meshed using triangular elements. Geometric nonlinearity was included in the model so that the FEA result captures the large deformation behavior more precisely. The elastic modulus used in the FEA was set to be 2230 MPa and was experimentally verified per ASTM D638 protocol.

## Data availability

All data presented in this work are available upon reasonable request from the corresponding authors.

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

## Acknowledgements

The authors would like to thank Logan Bekker for his assistance with projection microstereolithography. This work was performed under the auspices of the U.S. Department of Energy by Lawrence Livermore National Laboratory under Contract DE-AC52–07NA27344, and supported by Laboratory Directed Research and Development under grant 17-FS-007 & 19-ER-018. This work was additionally supported in part by AFOSR under award number FA9550–15–1–0321, by Prof. Hopkins' DOE-nominated Presidential Early Career Award for Scientists and Engineers under award number B620630, and by the NSF Graduate Research Fellowship award number DGE-1650604. The authors acknowledge program officer Byung "Les" Lee. LLNL-JRNL-751319.

## Author contributions

Y.S. designed the logic gates and performed the experimental measurements. A.J.P. and R.M.P. conceived the idea of functionally complete flexure-based logic gates. R.M.P., J.B.H. and A.J.P. improved the logic gate designs. S.C., L.A.S. and J.A.J. fabricated the mechanical logic gates. All authors discussed the results and wrote the paper. A.J.P. supervised the project.

## Additional information

**Competing interests:** The authors declare no competing interests.

