## [Peer Review File · Nature Communications]

Reviewers' comments:

Reviewer #1 (Remarks to the Author):

The authors present a design concept for mechanical logic gates fabricable across wide ranges of length scales by methods of additive manufacturing. The new design of various logic gates is introduced along with finite-element results demonstrating the mechanical functionality based on bistable elements. In addition, the authors present proof-of-concept experiments on selected fabricated logic gates on several scales (from macroscopic samples to micro-stereolithographic ones).

The particular design concept presented here is interesting and original to the best of this reviewer's knowledge, even though the concept of exploiting bistability either in a symmetric arrangement (as shown here) or in an asymmetric arrangement was demonstrated before. The manuscript is well written and the illustrations are instructive and support the conclusions.

Unfortunately, the scientific advances over existing concepts are incremental: the concept of using bistability for mechanical logic is well understood, the shown design is a modification of existing ones, the fabrication techniques are rather standard, the numerics are simple case studies using FEM software, and no new theory is presented. Moreover, no new science is reported rather than an engineering design concept. The paper further does not discuss or demonstrate how these logic gates are to be connected to achieve more complex computations. As a key conclusion, the logic gates are claimed to be reset easily and therefore can be reused (unlike prior designs that had challenges with resetting the gates); it is not clear to this reviewer how exactly the resetting is envisioned other than removing the horizontal precompression h (which implies that there needs to be active resetting indeed). Besides, the symmetric configuration makes it random to which of the two states the gates will switch upon compression, which may call for additional control mechanisms.

In conclusion, while the presented concepts appear sound (except for missing information about the resetting), they are more appropriate for an engineering journal rather than warranting publication in Nature Communications.

Reviewer #2 (Remarks to the Author):

This paper reports the development of mechanical logical gates that can be produced at a micrometer scale. A complete set of logical operators is proposed that do not require resetting and that can be scaled down to micrometer scale due to the compliant nature of the bistable constitutive elements. At the cost of a preloading step during fabrication, the two stable states of the bistable elements contain the same amount of potential energy.

Macro scale designs are presented and tested. The results match well with simulations. Of particular interest are the micro designs, that were produced with a combination of a holographic optical tweezer system for the manipulation of parts and a two-photon stereolithography system for their additive manufacturing. This combined system is able to assemble parts under preloading and is an important step to the actual realization of a reasonably sized mechanical computer.

While the treatise marks a clear contribution and appears technically sound, several issues remain:

- The phrase "perform digital calculations" is perhaps a little premature. While the gates can potentially be used to this end they cannot yet as they are still isolated and their mechanical connection is not trivial. Similarly, the claims can be phrased more precisely, i.e. that logical elements (not a computation approach) were proposed and produced at microscale.
- The Comsol modeling requires some more detail (type of element, geom nonlin, etc). This can be included in the supplementary material.

- It is unclear how the error bounds were determined (are they the result of multiple measurement, worst case, 1-sigma, etc). Also it is likely that the error bounds will be (much) worse in the microscale devices. Since they were produced the authors should give the reader an idea of their performance, e.g. the ratio of buckling forces in the microsystems relative to the optical tweezer force capability, magnitude and symmetry of nesting forces in the two stable equilibrium positions, etc. Such a treatise would also provide more dimensional detail.

Reviewer #3 (Remarks to the Author):

This is a well-written paper that describes a novel and interesting flexure-based design for mechanical logic gates that could be manufactured using established MEMS microlithography techniques and in principle can be combined to make more complex devices such as shift registers, memory cells, and the like. A method is demonstrated for fabricating micron-scale devices with a feature size as small as 600 nm. The citations of relevant recent literature appear complete, and there seems to be sufficient detail to allow replication of the work by another researcher.

The authors describe undesired buckling modes for the OR and AND gates and assert that insertion of a rigid crossbar with mechanical stops (Fig. S2(a)(b)) will fix the problem. Surface forces can be considerable at this scale and the large rigid body "C" appears to have larger surface area than the crossbar – have the authors modeled the crossbar fix to determine how much excess force would have to be applied to the larger rigid body to deform the smaller crossbar sufficiently to ruin the fix, either by bending the crossbar or by detaching the crossbar from the bottom of the larger rigid body by breaking the the small (~600 nm wide) connector? Have the authors estimated the force needed to separate two flat surfaces in contact (e.g., between the tops of A and B and the underside of C in Fig. 2(d), or between the bottom of C and the pedestal it contacts in Fig. 2(e) between Mode II&II and Mode IV) in a microscale device? The practicality of this design at very small scale depends upon these forces remaining modest in size compared to the forces needed to actuate the gates.

It would also be useful to see an estimate of: (1) the smallest gate size of this design that might be feasible to fabricate using known techniques, (2) the maximum speed (Hz) that these gates might be cycled, and (3) an estimate of energy dissipation per switching operation at relevant operating speeds (a significant metric in comparing the scalability of new gate designs).

Reviewers' comments:

Reviewer #1 (Remarks to the Author):

The authors present a design concept for mechanical logic gates fabricable across wide ranges of length scales by methods of additive manufacturing. The new design of various logic gates is introduced along with finite-element results demonstrating the mechanical functionality based on bistable elements. In addition, the authors present proof-of-concept experiments on selected fabricated logic gates on several scales (from macroscopic samples to micro-stereolithographic ones).

The particular design concept presented here is interesting and original to the best of this reviewer's knowledge, even though the concept of exploiting bistability either in a symmetric arrangement (as shown here) or in an asymmetric arrangement was demonstrated before. The manuscript is well written and the illustrations are instructive and support the conclusions.

Unfortunately, the scientific advances over existing concepts are incremental: the concept of using bistability for mechanical logic is well understood, the shown design is a modification of existing ones,

We thank the reviewer for their detailed comment. To our knowledge, there are no existing research papers or designs that utilize the mechanical bistability to perform functionally complete logic operations. We believe that we are the first research group to propose this “bi-stable flexure” design approach. We would be grateful if the reviewer could be more specific about these existing designs.

the fabrication techniques are rather standard,

While we do demonstrate three fabrication techniques, and fused deposition modeling is available commercially, the combination of 2 photon lithography and holographic optical tweezers is, to our knowledge, unique. Furthermore, we are the first group that utilizes this fabrication technique to fabricate micro-structures with embedded strain energy. Another paper by our group was recently submitted that describes the optical force and torque profiles on the microstructures of arbitrary shape to optimize the microfabrication process of the microsystems with embedded strain energy. Additional details of the fabrication method have been added to the Supplemental Information.

the numerics are simple case studies using FEM software, and no new theory is presented. Moreover, no new science is reported rather than an engineering design concept.

In this paper, we attempt to focus on the general design paradigm and working principles of such logic computations. This paper proposes a novel approach of designing mechanical digital computing systems, and the detailed discussion of the microfabrication capabilities and the performance of micro-fabricated logic gates will be provided in a later paper.

The paper further does not discuss or demonstrate how these logic gates are to be connected to achieve more complex computations. As a key conclusion, the logic gates are claimed to be reset easily and therefore can be reused (unlike prior designs that had challenges with resetting the gates); it is not clear to this reviewer how exactly the resetting is envisioned other than removing the horizontal precompression h (which implies that there needs to be active resetting indeed).

In the main article, we state that one of the main advantages of the proposed logic gate design is that it does not require resetting. Logic operations are performed in a continuous manner and the presence of the input(s) will immediately trigger the operation.

Besides, the symmetric configuration makes it random to which of the two states the gates will switch upon compression, which may call for additional control mechanisms.

The discussion about this point has been added to supplemental information, "*Logic Gate Designs*".

In conclusion, while the presented concepts appear sound (except for missing information about the resetting), they are more appropriate for an engineering journal rather than warranting publication in Nature Communications.

Reviewer #2 (Remarks to the Author):

This paper reports the development of mechanical logical gates that can be produced at a micrometer scale. A complete set of logical operators is proposed that do not require resetting and that can be scaled down to micrometer scale due to the compliant nature of the bistable constitutive elements. At the cost of a preloading step during fabrication, the two stable states of the bistable elements contain the same amount of potential energy.

Macro scale designs are presented and tested. The results match well with simulations. Of particular interest are the micro designs, that were produced with a combination of a holographic optical tweezer system for the manipulation of parts and a two-photon stereolithography system for their additive manufacturing. This combined system is able to assemble parts under preloading and is an important step to the actual realization of a reasonably sized mechanical computer.

While the treatise marks a clear contribution and appears technically sound,

several issues remain:

- The phrase "perform digital calculations" is perhaps a little premature. While the gates can potentially be used to this end they cannot yet as they are still isolated and their mechanical connection is not trivial. Similarly, the claims can be phrased more precisely, i.e. that logical elements (not a computation approach) were proposed and produced at microscale.

We have made changes (line 31) in the paper to reflect the capability of the design more precisely. Namely, replaced "digital" with "Boolean"

- The Comsol modeling requires some more detail (type of element, geom nonlin, etc). This can be included in the supplementary material.

A section about the FEA model has been added to the Methods section.

- It is unclear how the error bounds were determined (are they the result of multiple measurement, worst case, 1-sigma, etc).

The error bound standard has been added to the caption of Figure S5 in supplemental material.

Also it is likely that the error bounds will be (much) worse in the microscale devices. Since they were produced the authors should give the reader an idea of their performance, e.g. the ratio of buckling forces in the microsystems relative to the optical tweezer force capability, magnitude and symmetry of nesting forces in the two stable equilibrium positions, etc. Such a treatise would also provide more dimensional detail.

Ideally (according to simulation), the forces scale proportionally with the cube of any characteristic length for all logic gate designs. For instance, the required switching force for a 10um x 10um x 1um NOR gate will be on the order of piconewtons, and the pre-loading force will be one magnitude larger than the switching force. Preliminary experimental results showed that the optical force on a micron-sized feature can reach up to 60pN, but these results need further analysis and verification.

We are reluctant to discuss these numbers in the paper because these results are generated by faithfully scaling the proposed design without further experimental verification. We are concerned that providing these numbers in the paper will mislead the readers to believe that these numbers are verified experimentally. We have thus decided to show the fabrication example of a "bit" cell (Figure 3e) to demonstrate the optical tweezer force capability qualitatively instead of quantitatively.

The proposed fabrication technique was recently developed by our team at UCLA. The optical trapping force and the micro-fabrication resolution are highly dependent on the geometric shape and photo-polymer material. We are currently working on a simulation tool that describes the optical force and torque profiles on the microstructures of arbitrary shape to optimize the microfabrication process of the microsystems with embedded strain energy. As we learn more about the microfabrication process, redesign of certain features of the logic gate design

may be necessary. A detailed discussion of the microfabrication capabilities will be provided in a later paper.

Reviewer #3 (Remarks to the Author):

This is a well-written paper that describes a novel and interesting flexure-based design for mechanical logic gates that could be manufactured using established MEMS microlithography techniques and in principle can be combined to make more complex devices such as shift registers, memory cells, and the like. A method is demonstrated for fabricating micron-scale devices with a feature size as small as 600 nm. The citations of relevant recent literature appear complete, and there seems to be sufficient detail to allow replication of the work by another researcher.

The authors describe undesired buckling modes for the OR and AND gates and assert that insertion of a rigid crossbar with mechanical stops (Fig. S2(a)(b)) will fix the problem. Surface forces can be considerable at this scale and the large rigid body "C" appears to have larger surface area than the crossbar – have the authors modeled the crossbar fix to determine how much excess force would have to be applied to the larger rigid body to deform the smaller crossbar sufficiently to ruin the fix, either by bending the crossbar or by detaching the crossbar from the bottom of the larger rigid body by breaking the the small (~600 nm wide) connector?

The undesired buckling modes will only occur if the large rigid body "C" is subject to a small disturbance near its equilibrium position (Figure 1c and near 0 position). when "C" transitioning to a different logic state. The function of the crossbar is not to force the rigid body "C" from its stable position (the undesired position) back to its desired position, but rather to prevent the rigid body "C" from going towards the undesired position during the transition by introducing more resistance (higher energy barrier) for on the side of "undesired position".

Therefore, the additional crossbar does not need to withstand the large force that is required for pushing the rigid body "C" back into the correct position, but only needs to withstand the minor disturbance that may occur during the transition process.

Have the authors estimated the force needed to separate two flat surfaces in contact (e.g., between the tops of A and B and the underside of C in Fig. 2(d), or between the bottom of C and the pedestal it contacts in Fig. 2(e) between Mode II&II and Mode IV) in a microscale device? The practicality of this design at very small scale depends upon these forces remaining modest in size compared to the forces needed to actuate the gates.

The stiction of two flat surfaces may be reduced and eliminated by adding anti-stiction bumps on those flat surfaces. In fact, the proposed fabrication technique (2PS-HOT) was just recently developed by our team at UCLA, and as we learn more about the micro-fabrication process, redesigning certain geometric features of the logic gates is certainly expected. However, in this paper we attempt to

focus on the general design paradigm and working principles of such logic computations. A detailed discussion of the microfabrication capabilities and the performance of micro-fabricated logic gates will be provided in a later paper.

It would also be useful to see an estimate of: (1) the smallest gate size of this design that might be feasible to fabricate using known techniques

This is discussed in supplementary information (line 68).

(2) the maximum speed (Hz) that these gates might be cycled,

The speed largely depends on the material and the flexure design. As a rough estimate, logic gates at micron-scale may operate at Mhz rate. We feel that such estimations should only be provided if they are supported by experimental data. Our future work will provide the information about the micro-fabrication and testing of logic gates at this scale.

and (3) an estimate of energy dissipation per switching operation at relevant operating speeds (a significant metric in comparing the scalability of new gate designs).

In order to make such an estimate, we will need to define the scale of the logic gates, the operating environment, material properties etc. according to application scenarios. These factors may only be reasonably determined after obtaining full knowledge of the microfabrication capabilities. These are not within the scope of this paper but will be addressed in our future work.

Reviewers' comments:

Reviewer #1 (Remarks to the Author):

The authors have responded to the points raised previously by this reviewer. The added information in the manuscript is appreciated, yet the key point of missing novelty is not entirely clarified. To be specific:

The authors cite e.g. Raney et al. (PNAS 2016) who did demonstrate bi-stability-based flexural logic gates, so stating that "we believe that we are the first research group to propose this 'bi-stable flexure' design approach" unfortunately does not apply. Using bistability and flexural deformation has been exploited by Raney and co-authors in multiple papers; in the cited one (PNAS 2016) AND and OR gates were demonstrated. It is correct that not all possible logic operations were shown there, but the general design strategy used here was outlined there. Besides, the original idea of using flexural bistability for logic goes back to at least Merkle (Nanotechnology 4 (1993), p. 114, "Two Types of Mechanical Reversible Logic").

The authors state that "This paper proposes a novel approach of designing mechanical digital computing systems, and the detailed discussion of the microfabrication capabilities and the performance of micro-fabricated logic gates will be provided in a later paper". Hence the focus and major advances here are not fabrication or modeling. Since the key point here is on the design of the logic operations, which - given the above comment - are similar to those proposed previously, the novelty is limited. If larger systems of connected gates had been shown, that would have been a real novelty but demonstrating isolated logic gates based on a previously published physical phenomenon (bistability in flexural elements) is interesting but, in this reviewer's opinion, not necessarily Nature Communications level.

Reviewer #2 (Remarks to the Author):

Most of the comments have been addressed. The additional information on the FEA modelling is appreciated. The frank report on the preliminary nature of the microdevices is also appreciated, although it would still be good to indicate the potential performance (several aspects) as this would support the relevance of the work.

Reviewer #3 (Remarks to the Author):

The manuscript is acceptable for publication.

Reviewer #1 (Remarks to the Author):

The authors have responded to the points raised previously by this reviewer. The added information in the manuscript is appreciated, yet the key point of missing novelty is not entirely clarified. To be specific:

The authors cite e.g. Raney et al. (PNAS 2016) who did demonstrate bi-stability-based flexural logic gates, so stating that "we believe that we are the first research group to propose this 'bi-stable flexure' design approach" unfortunately does not apply. Using bistability and flexural deformation has been exploited by Raney and co-authors in multiple papers; in the cited one (PNAS 2016) AND and OR gates were demonstrated. It is correct that not all possible logic operations were shown there, but the general design strategy used here was outlined there. Besides, the original idea of using flexural bistability for logic goes back to at least Merkle (Nanotechnology 4 (1993), p. 114, "Two Types of Mechanical Reversible Logic").

The authors state that "This paper proposes a novel approach of designing mechanical digital computing systems, and the detailed discussion of the microfabrication capabilities and the performance of micro-fabricated logic gates will be provided in a later paper". Hence the focus and major advances here are not fabrication or modeling. Since the key point here is on the design of the logic operations, which - given the above comment - are similar to those proposed previously, the novelty is limited. If larger systems of connected gates had been shown, that would have been a real novelty but demonstrating isolated logic gates based on a previously published physical phenomenon (bistability in flexural elements) is interesting but, in this reviewer's opinion, not necessarily Nature Communications level.

We are glad that the reviewer mentioned these papers. We discussed these papers in the introductory section but could not make a detailed comment about these papers due to the length limitation.

Raney et al. proposed an approach to propagate mechanical signals rather than perform logic operations, as indicated by the title of the paper, *Stable propagation of mechanical signals in soft media using stored elastic energy*. The key difference is whether it is possible to generate a NOT gate design. Any functionally complete set of logic operators is required to have the capability to invert an input signal, i.e., to switch 0 to 1 and 1 to 0. However, it is fundamentally not possible to perform such inversion operation based on the design scheme proposed in this paper. Their approach utilizes soft materials to construct a non-symmetric bi-stable element, and each switching element releases the previously stored energy to trigger the next element. Between the two logic-inverting processes: “0 to 1” and “1 to 0”, there must be one process that is energy-absorbing rather than energy-releasing. Such energy-absorbing operation cannot be performed without an external energy source besides the signal itself. This is a fundamental limit of this design that cannot be resolved by any incremental improvement. Therefore, this design strategy cannot be used to design functionally complete digital computing systems. The authors did not make such claims either.

Merkle proposed the original idea of the mechanical reversible logic utilizing symmetric bi-stability. We consider this paper to be a theoretical basis of mechanical logic gate designs, but this paper proposed theoretical concepts rather than a feasible design approach. Practical issues like the physical size, geometric constraints, interconnection of logic gates, material properties and feasible fabrication approaches are not discussed. No experimental results or even FEA data was shown.

We apologize for oversimplifying our contributions in the last response. It was our intention to convey that we believe we are the first research group to propose a practical 'bi-stable flexure' design approach to design logic gates that perform functionally complete logic operations. Raney's approach cannot be applied to design a functional-complete set of logic gates and Merkle proposed a theoretical concept instead of a practical design approach. To be more accurate, here is what we believe to be our main contribution:

We proposed a bi-stable-flexure-based approach to design functionally complete sets of logic gates that can be microfabricated. Such logic gates can be interconnected with each other to construct a larger logic operation unit that performs continuous logic operation with no need for “resets”.

These claims are demonstrated by constructing a NAND gate. The design of the NAND gate is not independently conceived, but rather constructed by interconnecting two NOT gates and an OR gate. It is verified by FEA and experiments to operate as designed. An even larger logic operation unit may also be constructed similarly. A fabrication approach is proposed for the microfabrication of such logic systems and initial results indicate it is feasible for fabrication more complex logic systems.

Reviewer #2 (Remarks to the Author):

Most of the comments have been addressed. The additional information on the FEA modelling is appreciated. The frank report on the preliminary nature of the microdevices is also appreciated, although it would still be good to indicate the potential performance (several aspects) as this would support the relevance of the work.

We have added some more information about the current capability of the micro-fabrication system and the logic gate performance as suggested.

Specifically, in the supplemental information, we added: "The current system has a minimum micro-fabrication resolution of 800nm and can generate optical trapping forces of up to 50pN, which in theory is capable of fabricating a logic gate of 100um size. A rough estimation from first principles suggests that logic gates at this scale can perform logic operations at Mhz speed. This estimation will be further investigated and verified in our future work."

Reviewer #3 (Remarks to the Author):

The manuscript is acceptable for publication.

We thank the reviewer for their time and effort in reviewing our manuscript.